# Development and accuracy evaluation of a new loop-mediated isothermal amplification assay targeting the HSP70 gene for the diagnosis of cutaneous leishmaniasis

**Arthur Ribeiro Cheloni Soares, Verônica Cardoso Santos de Faria, Daniel Moreira de Avelar**⦿*

Instituto René Rachou, Fundação Oswaldo Cruz, Grupo de Pesquisa Clínica e Políticas Públicas em Doenças Infecciosas e Parasitárias, CEP: 30190–002, Belo Horizonte, Minas Gerais, Brazil

* daniel.avelar@fiocruz.br

**Data Availability Statement:** All relevant data are within the manuscript and its Supporting Information files.

## Abstract

Cutaneous leishmaniasis (CL) is a global public health problem caused by species on the genus *Leishmania* and is the most prevalent clinical form of leishmaniasis. The aim of this study was to develop a new LAMP assay for *Leishmania* sp. based on HSP70 gene and evaluate it clinically for molecular diagnosis of CL. The study was carried out in the following stages: i) design of primers based on HSP70 gene of *Leishmania* sp.; ii) evaluation of detection limit and analytical specificity; iii) estimation of the accuracy of LAMP-Leish/HSP70 assay for diagnosing CL. A total of 100 skin biopsy samples from patients, comprising 60 CL cases and 40 non-cases, were analyzed in this study. One LAMP assay using HSP70 gene as molecular target were standardized, and the observed detection limit was 100fg of *L. braziliensis* purified DNA. The LAMP-Leish/HSP70 assay was specific for *Leishmania* spp. The LAMP-Leish/HSP70 assay showed an accuracy of 92%, and positivity rates were not affected by lesion onset time or parasite load. This novel LAMP assay targeting the HSP70 gene of *Leishmania* sp. has the potential to be a useful tool to integrate into routine diagnosis for suspected cases of CL.

## Introduction

Tegumentary leishmaniasis (TL) is a neglected tropical disease that primarily affects low-income populations in 90 endemic countries [1]. In the Americas, more than 1,100,000 cases of TL were reported between 2001 and 2021 [2]. The TL disease can present in two main clinical forms: cutaneous leishmaniasis (localized, disseminated or diffuse) and mucosal or mucocutaneous leishmaniasis (MCL) [3].

The laboratory tests currently available for the diagnosis of TL do not provide sufficient accuracy to be considered the gold standard. Laboratory diagnosis of this disease still presents a challenge with little investment in the development of laboratory tests by major diagnostic industry segments [4,5]. The diagnosis of CL has been based on the association of clinical

**Funding:** This study was supported by Fundação de Amparo à Pesquisa do Estado de Minas Gerais (FAPEMIG) in the form of a grant [APQ-00802-20], the Conselho Nacional de Desenvolvimento Científico e Tecnológico (CNPq), Brasil in the form of a grant [408146/2021-4], and by FAPEMIG in the form of a scholarship [1324553/PAPG 2017-20] to ARCS.

**Competing interests:** The authors have declared that no competing interests exist.

characteristics, epidemiological data, and results of laboratory tests [6]. The direct search for amastigotes in lesions remains the main test in laboratories of primary health care centers, despite its variable sensitivity [6,7]. The effectiveness of isolation in culture is also low and very variable, with studies reporting 10% and 50% sensitivity, with the technique used, the scarcity of the parasite in the lesions and the frequent contamination of the culture medium with fungi and bacteria pointed as possible responsible for this variation [8,9].

Immunological tests, such as the *Leishmania* skin test and serology-based assays, exhibit variable sensitivity and/or specificity, depending on the antigen used, the patient's immunological status, and the possibility of cross-reactions with other diseases [7,10]. Among immunological tests, the CL Detect™ Rapid Test, that detects the peroxidoxin antigen produced by *Leishmania* amastigotes in skin lesions, has been evaluated in some endemic countries (31.3%-65.4% sensitivity when compared to microscopy and PCR) [11–14].

Molecular diagnostic techniques targeting various *Leishmania* genes have been developed for TL diagnosis [15] and have shown remarkable accuracy (meta-analyses study with pooled sensitivity and specificity of 95% and 97%, respectively) ([16]), even using minimally invasive sampling methods [17–19]. Among molecular techniques, PCR platforms (conventional PCR and real time PCR) are the most used methods for TL diagnosis. However, these molecular tools require advanced laboratory facilities, making it difficult to implement in remote areas from high TL-burden countries [17].

The loop-mediated isothermal amplification (LAMP) method is a simple rapid diagnostic tool for nucleic acid detection and diagnosis of infectious diseases. The main advantages of LAMP include its high accuracy, robustness, fast reaction, no need for thermal cycling, and that the product can be detected through turbidity, fluorescence or change of color [20]. LAMP assays for CL were evaluated in several endemic settings and demonstrated sensitivity and specificity values ranging from 82.6% to 100% and 42.9% to 100%, respectively [12,13,20–28]. In Brazil, only one accuracy study of LAMP assay (18S) for the diagnosis of TL was carried out, and a sensitivity of 86% was obtained with biopsy samples and 82.5% with swab samples. The specificity obtained was 92.6% and 100%, with biopsy and swab samples, respectively [29]. Though cross reaction of the 18S LAMP assay was observed with *Trypanosoma brucei* and *T. cruzi* [28].

A 2019 target product profile (TPP) for a dermal leishmaniasis point-of-care test calls for a rapid, simple, and robust test suitable for resource-limited settings. Minimal requirements include genus-specific detection of active localized CL with over 90% specificity and at least 95% sensitivity. Results should be available in under 1 hour through visual reading or using a simple reading device [30]. Diagnostics tests based on LAMP can fulfill these criteria, making them promising candidates for meeting the requirements outlined in the TPP. This study reports the development and evaluation of a novel LAMP assay for detection of HSP70 gene of *Leishmania* spp. The target was selected because it is commonly used for molecular diagnosis of Leishmaniasis, including *Leishmania* species identification. This target presents both conserved and polymorphic regions, making it useful for differentiating a wide range of species from different geographical origins, especially species causing American TL. In this study, the accuracy of a LAMP- Leish/HSP70 assay to diagnose CL was evaluated for the first time.

## Materials and methods

### Alignment of *Leishmania* spp. genome sequences and primer design

*Leishmania* spp. genome sequences of HSP70 gene were downloaded from NCBI GenBank (https://www.ncbi.nlm.nih.gov/genbank/) database. A comparative analysis was made by aligning the genome sequences using VISTA (http://genome.lbl.gov/vista/index.shtml). The

conserved genus-specific regions of *Leishmania* were selected for the design of primers. All LAMP primers sets, each containing four primers, were designed using PrimerExplorer V5 program (https://primerexplorer.jp/e). Additionally, the specificity of LAMP primers was confirmed through BLAST searches against the NCBI database. The forward outer primer (F3) and backward outer primer (B3) of all sets were desalted, while the forward inner primer (FIP) and backward inner primer (BIP) were HPLC-purified (IDT, Iowa, EUA).

## Preparation of DNA reference

Genomic DNA was obtained from the following *Leishmania* reference strains: *L. (Leishmania) amazonensis* (IFLA/BR/1967/PH-8), *L. (Viannia) braziliensis* (MHOM/BR/75/M2903); *L. (L.) donovani* (MHOM/ET/67/HU3), *L. (V.) guyanensis* (MHOM/BR/1975/M4147), *L. (L.) infantum* (MHOM/BR/74/PP75); *L. (V.) lainsoni* (MHOM/BR/81/M6426); *L. lindenbergi* (MHOM/BR/1996/M15733); *L. major* (MHOM/SU/73/5-ASKH); *L. mexicana* (MNYC/BZ/62/M379); *L. naiffi* (MDAS/BR/1979/M5533); *L. panamensis* (MHOM/PA/71/LS94); and *L. (V.) shawi* (MCEB/BR/1984/M8408). Analytical specificity was assessed by testing DNA samples from *T. cruzi* (Y strain), *Toxoplasma gondii* (ME49 strain), *Schistosoma mansoni* (BH strain), *Plasmodium* sp.; *Chromobacterium* sp.; *Sporothrix schenckii*. DNA was extracted using PureLink™ Genomic DNA Mini Kit (Invitrogen/Thermo Fisher Scientific, USA) according to the manufacturer's instructions. The concentration of genomic DNAs was determined using a Nano-Drop ONE (Thermo Fischer Scientific Inc., Waltham, MA). The purity of obtained DNA was verified by analyzing the A260/280 and 260/230 absorbance ratios.

## Standardization of *Leishmania* LAMP assay

An evaluation of the effects of different concentrations of internal and external primers (8:2mM and 16:2mM), Bst 2.0 –WarmStart DNA polymerase (New England Biolabs®) (4 to 8 U) and the reaction time (continuous analysis to 90 min) were carried out to optimize the LAMP asssay. Other reagents were used in following concentrations: 1 mM DNTPs, 0.8 M betaine, 20 mM Tris-HCl (pH 8.8), 10 mM kCL, 10 mM $[NH_4]_2SO_4$, 8 mM $MgSO_4$, 1% Tween 20, and 2 µL (200 pg) of *L. braziliensis* DNA (MHOM/BR/75/M2903). On the inner side of tube's cover, were addicted 1 µL of SYBR Green I 10.000X/DMSO (Invitrogen®), diluted at 1:10. Amplification reactions for standardization were carried out in 25 µL volume using a water bath device (Lindberg/Blue M, Thermo Fischer Scientific). The temperature of 65˚ C ± 1˚ C was selected based on the optimal temperature of BST DNA polymerase. At the end of reaction, each tube was briefly centrifuged to allow mixing of the SYBR Green I dye and the amplified product for visual analysis of the results [31]. The negative samples turned orange while the positives turned green due to the intercalating dye's reaction with DNA. For further confirmation, 4 µL of the LAMP products were visualized after electrophoresis on a 6% polyacrylamide gel and silver stained. The raw images were provided as S1 Raw images. All tests were made in duplicate.

## Detection limit and analytical specificity

The limit of detection (LOD) for each LAMP primer set was determined by using serial dilutions of purified genomic DNA from *L. braziliensis*, ranging from 1 ng to 1 fg. The experiments were performed in duplicate, and nuclease-free water was used as negative control. The analytical specificity of the LAMP assays was tested against human genomic DNA extracted from buffy coat and genomic DNA anteriorly cited in "Preparation of DNA reference" section.

## CL diagnosis: Accuracy study design, parasite load quantification and ethics statement

The clinical evaluation of index test (LAMP-Leish/HSP70 assay) was performed using the protocol defined after the standardization. The present study followed the recommendations of the STARD 2015 [32]. The accuracy panel-based study of LAMP assay was performed with genomic DNA extracted from skin biopsy samples obtained from patients with a suspected clinical presentation of CL and had attended the Leishmaniasis Reference Center of Instituto René Rachou, Fundação Oswaldo Cruz. The samples were coded, and the execution was performed blindly, without clinical information or results from other laboratorial tests. All individuals were prospectively recruited and a written informed consent was obtained from each participant or from their legal guardian for underage participants. The recruitment period occurred from 2017 to 2019. The Ethical Research Committee of the Research Center René Rachou/FIOCRUZ approved this study (FIOCRUZ/CAAE: 44545315.7.0000.5091).

DNA was extracted using PureLink™ Genomic DNA Mini Kit (Invitrogen/Thermo Fisher Scientific, USA) according to the manufacturer's instructions. The ratios of purity and concentrations of genomic DNA were determined using a NanoDrop ONE system (Thermo Fischer Scientific Inc., Waltham, MA).

The minimum sample size required for this study was estimated based on a CL prevalence of 60.8% for the target population (patients attending at the reference center previous cited). The sample size calculation was based on tables and formula proposed by [33]. The number of controls was first estimated based on an expected specificity of 95% [10,17]. As the prevalence in the target population > 50%, the number of controls was first estimated considering a minimal acceptable lower confidence limit of 0.75 (1 –α), the inclusion of at least 34 controls was recommended. The minimal number of CL cases were 53, based on the following formula proposed by the same authors: number of controls = number of cases [(1- disease prevalence) / disease prevalence].

The inclusion criteria included both clinical and laboratory diagnosis. The patient must have presented a cutaneous ulcer, with a granular bottom and infiltrated edges in a frame. The kDNA-qPCR [29] was used as the reference standard for CL diagnosis. All samples of the non-CL group were negative in kDNA-qPCR. Patients undergoing treatment or previously treated for leishmaniasis ≤ 1 year before recruitment were excluded from the study.

The kDNA-qPCR was performed on all biopsy samples by the detection system with non-specific intercalating dye SYBR® Green (Applied Biossytems®, CA, USA) using the StepOnePlusTM Real-Time PCR System (Applied Biosystems®, CA, USA). The kDNA-qPCR protocol followed the description by [34]. In the present study, the kDNA-qPCR protocol used 30 cycles. The following primers were used: 150 forward primer: 5' (C/G) (C/G) (G/C) CC (C/A) CTA T(T/A)T TAC ACC AAC CCC3' and 152 reverse primer: 5' GGG GAGGGG CGT TCT GCG AA3'[35,36]. The reaction was carried out in a total volume of 25 μL containing 1.0 μL of each primer at a concentration of 10 pmols/μL, 12,5 μL of SYBR® Green, 5.5 μL of deionized water and 5.0 μL of DNA (10 ng/μL). The threshold of detection, baseline and melting curve were automatically determined using StepOne™ Software v.2.1. Results were obtained evaluating the following variables: the melting curve and quantification of the number of DNA copies. The standard curve was constructed using triplicate samples of *L. braziliensis* DNA (R2 0.99, efficiency 102.5%, slope -3.26), where 83.15 fg was considered equivalent to one parasite [37] The amount of *Leishmania* parasites was calculated following the calculation (parasite DNA equivalents per reaction / amount of tissues DNA per reaction) x $10^3$, expressed as *Leishmania* parasites per microgram of tissue DNA [38].

## Statistical analysis

The results of kDNA-qPCR on biopsy samples were regarded as the gold standard for molecular CL diagnosis. Sensitivity ($Se$), specificity ($Sp$), and diagnostic accuracy ($Acc$) were calculated using a two-by-two contingency table with exact binomial statistics, at a 95% confidence interval (95% CI). The results of LAMP-Leish/HSP70 assay were compared with parasitological tests performed in all patients (direct microscopy examination and aspirate of culture). Differences in $Se$, $Sp$, and $Acc$ were compared using McNemar's test. The interobserver reproducibility of LAMP assay results were assessed using the kappa index, following Landis and Koch (1977) criteria: <0, no agreement; 0–0.2, slight agreement; 0.2–0.4, fair agreement, 0.4–0.6, moderate agreement; 0.6–0.8, substantial agreement; 0.8–1, almost perfect agreement. Parasite load was categorized as "low" ($\leq$ 10 parasites/µg tissue) or "high" (> 10 parasites/µg tissue) based on distribution data assessed by the Kolmogorov-Smirnov test [39]. It was then analyzed in relation to the duration of the lesion ($\leq$ or > 3 months) in CL patients using the Mann-Whitney U test [37]. The non-parametric Mann-Whitney test was employed for comparison between positivity of LAMP assays and the following variables: time of duration of lesion and parasite load. Statistical analysis was performed using the Medcalc Software (Medcalc Software, Ostend, Belgium).

## Results

### Primer design

Five candidate primer sets were designed and after BLAST analysis, only one useful LAMP primers set was selected. The sequences of the oligonucleotides are provided in Table 1.

### Standardization of LAMP assay

The results indicated that the optimal ratio concentration between internal and external primers should be 16:2 mM, and the maximum amplification was achieved with 8U of Bst DNA polymerase. The reaction time of LAMP assay was established as 60 min (Fig 1). The final optimized LAMP assay conditions included incubation at 65° C. The reaction mixture (25µL) consisted of 1.6 µM each of FIP and BIP primers, 0.2 µM each of F3 and B3 primers, 8 U of 2.0 Bst-WarmStart DNA polymerase (New England BioLabs); 1 mM deoxynucleoside triphosphates, 0.8 M betaine, 20 mM Tris-HCL (pH 8.8), 10 mM KCl, 10 mM [NH$_4$]$_2$SO$_4$, 8 mM MgSO$_4$, 1% Tween 20, and 2 µL of template DNA.

### Detection limit (LOD) and specificity of the LAMP assays

The LODs of the HSP70-LAMP assay was 100 fg of *L. braziliensis* purified DNA (Fig 2). In the evaluation of analytical specificity, the LAMP-Leish/HSP70 assay showed specific positive results only for *Leishmania* spp (Fig 3).

**Table 1. LAMP primer sets used in this study.**

| Primers* | Sequence 5'– 3' | (bp) |
|---|---|---|
| HSP70_F3 | CTGCTGGACGTGACGC | 16 |
| HSP70_B3 | CGAGTGGCAGTCCTTCGT | 18 |
| HSP70_FIP | TGTTGCGCTTGATCAGCGCC−CGCTGACGCTGGGCATT | 37 |
| HSP70_BIP | GATCCCGACCAAGAAGAGCCAG−TCGTACACCTGGATGTGCA | 41 |

*F3, forward outer primer; B3, backward outer primer; FIP, forward inner primer; BIP, backward inner primer; bp, base pairs.

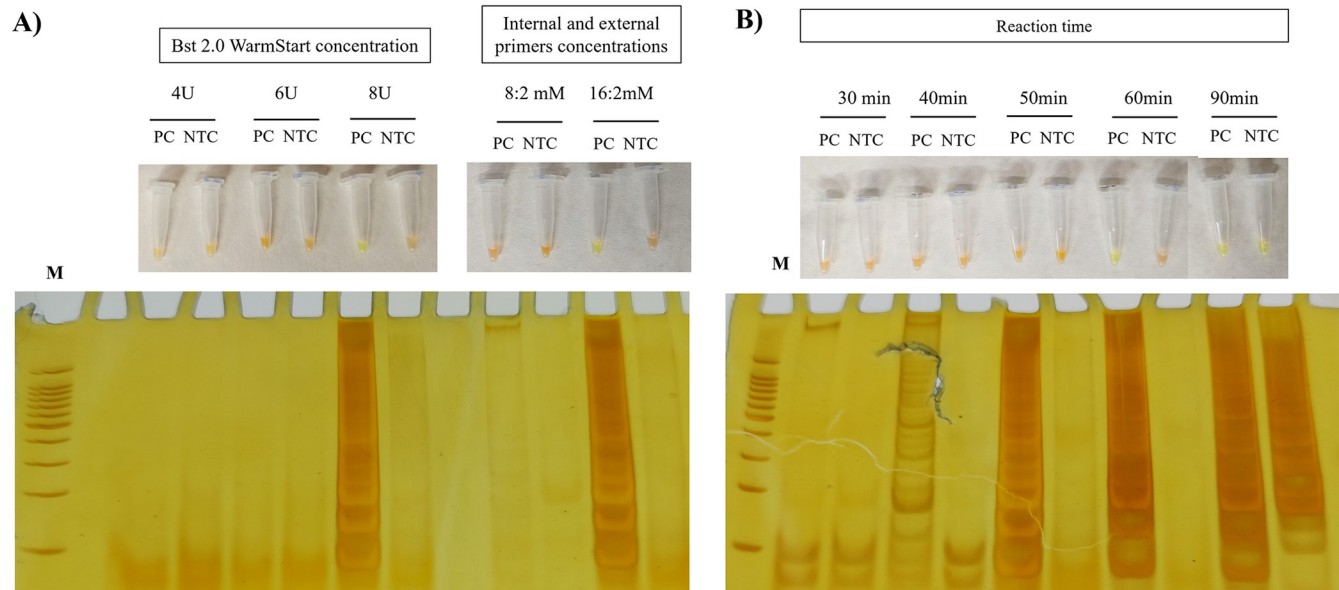

**Fig 1. Optimization of reagent concentrations and conditions of the HSP70 LAMP assay.** A) Bst 2.0 WarmStart DNA polymerase (ranging from 4 to 8U) and two concentrations of internal and external primers (8:2mM and 16:2mM); B) incubation time ranging from 30 to 90 min. The HSP70 LAMP results were observed by visual color change of the products in the reaction tube (orange color = negative reaction; and greenish yellow color = positive reaction) and by gel electrophoresis. Legends: PC: Positive control; NTC (non-template control): Water; M: Molecular weight marker: 100bp.

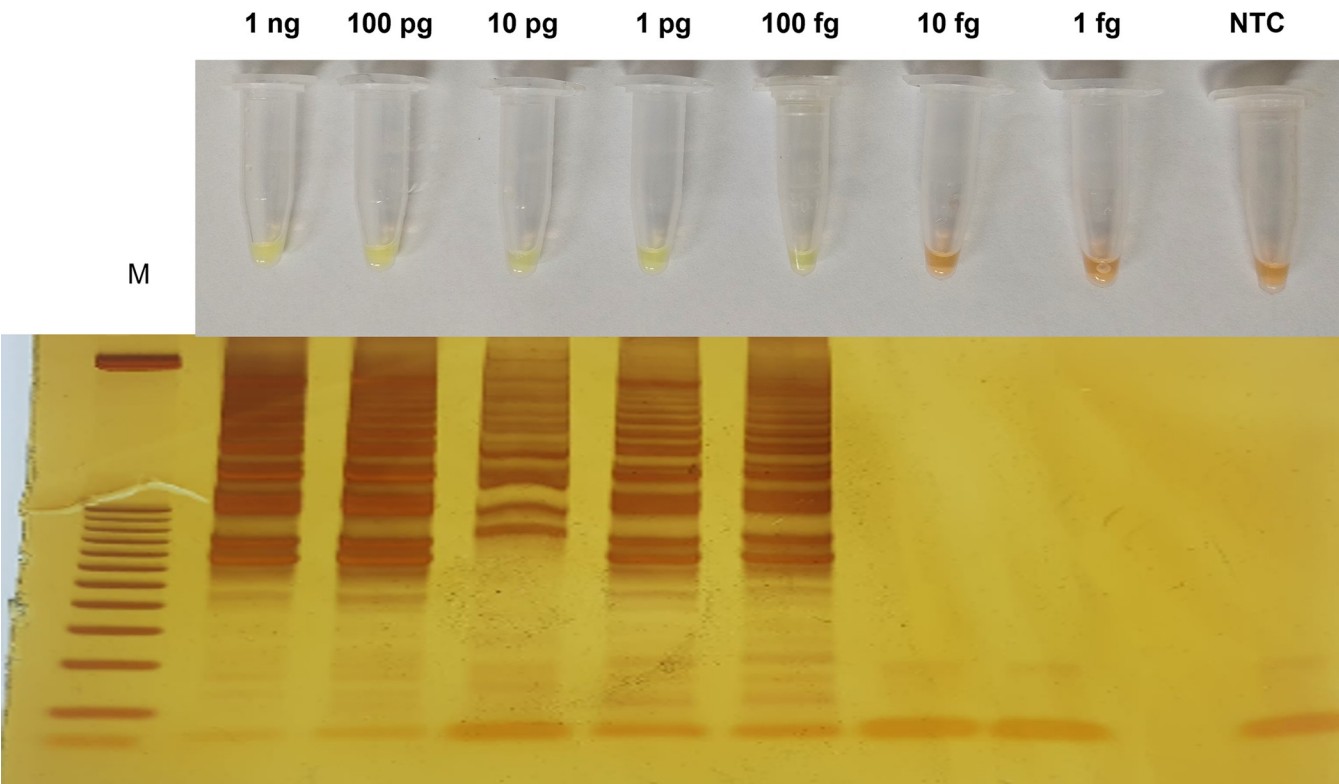

**Fig 2. Detection limit of LAMP-Leish/HSP70 assay determined based on serial dilutions of purified genomic DNA from *L. braziliensis*, ranging from 1 ng to 1 fg.** The LAMP results were observed by visual color change of the products in the reaction tube (orange color = negative reaction; and greenish yellow color = positive reaction) and by gel electrophoresis. Legends: PC: Positive control; NTC (non-template control): Water; M: Molecular weight marker: 100bp.

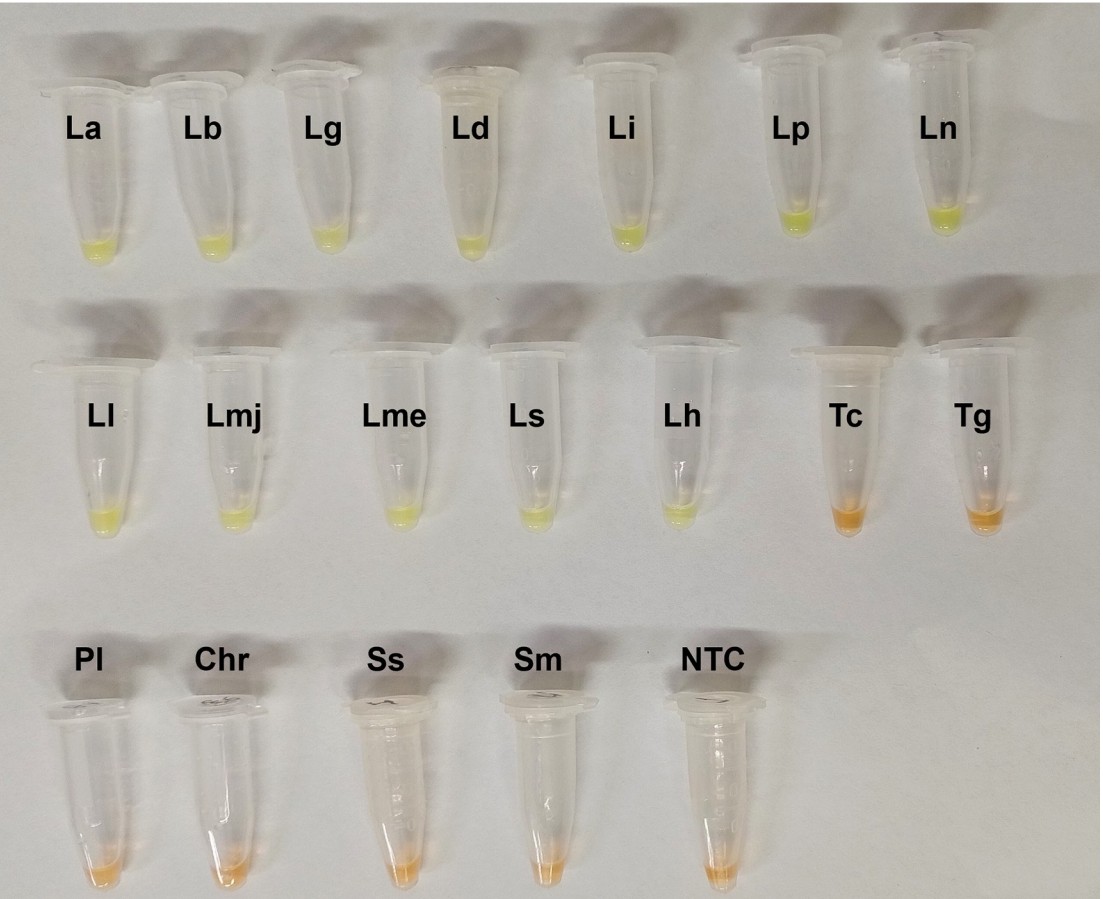

**Fig 3. Analytical specificity of *Leishmania* HSP70-LAMP assay.** The LAMP results were observed by visual color change of the products in the reaction tube (orange color = negative reaction; and greenish yellow color = positive reaction). Legends: PC: Positive control; NTC (non-template control): Water; La: *L. amazonensis;* Lb: *L. braziliensis;* Lg: *L. guyanensis;* Ld: *L. donovani;* Li: *L. infantum;* Lp: *L. panamensis;* Lh: *L. hertigi;* Ln: *L. naiffi;* Ls: *L. shawi;* Ll: *L. lindenbergi;* Lme: *L. mexicana;* Lmj: *L. major;* Tc: *Trypanosoma cruzi;* Tg: *Toxoplasma gondii;* Pl: *Plasmodium* sp; Chr: *Chromobacterium* sp.; Ss: *Sporothrix schenckii;* Sm: *Schistosoma mansoni.*

## Accuracy study of LAMP-Leish/HSP70 assay and comparison with parasitological tests

The accuracy study of LAMP-Leish/HSP70 assay was performed on DNA extracted of 100 skin biopsy samples, including 60 CL cases and 40 non-cases. Demographic and clinical data of all patients with CL are summarized in Table 2. Thirty-one out of 60 CL cases tested positive in the parasitological diagnosis (direct microscopy examination of biopsy samples and/or parasite culture). The number of lesions among patients with CL ranged from 1 to 15 lesions, and 55 patients presented the localized cutaneous form, two with disseminated lesions, and three with mucocutaneous leishmaniasis. No patient included in the study had HIV/*Leishmania* co-infection. The main demographic and clinical characteristics of all participant patients and the results of diagnostic tests are available in S1 Table.

The performance of the techniques was evaluated using qPCR as the gold standard. The LAMP-Leish/HSP70 assay demonstrated a *Se* of 86.7% (95% CI: 75,4 to 94,1), *Sp* of 100% (95% CI: 91.2 to 100), and an overall *Acc* of 92% (95% CI: 84.8 to 96.5). Only thirteen (21.7%) patients with CL were positive in the direct microscopy test. There was statistically significant

**Table 2. Demographic and clinical data of CL cases included in this study.**

| Characteristics | CL cases |
|---|---|
| Age, median (range), years | 60 (5–79) |
| Gender, male vs. females % | 68.3 x 31.7 |
| Duration of lesion, median (range), months | 4 (1–70) |
| **Location of lesion (%)** | |
| Upper limbs | 31.6 |
| Lower limbs | 51.7 |
| Thorax | 10 |
| Head | 6.7 |
| Type of lesion (%) | (96.7) ulcers / (3.3) plaques |

difference between *Se* rates presented by LAMP-Leish/HSP70 and direct microscopy examination and aspirate of culture (McNemar's test, $p \leq 0.05$) (Table 3). Total agreement was observed in the test results between observers (K = 1.0).

The median parasite load in lesions of CL cases was 2.33 parasites/µg of tissue DNA. Recent lesions ($\leq 3$ months) and lesions with a longer evolution time ($> 3$ months) showed the following median and interquartile range values, respectively: 5.3 (0.6–36.2) and 1.67 (0.43–8.9). No statistically significant difference was found between the parasite loads of the two groups (U Mann-Whitney test, p = 0.31) (raw data available in S1 Table). The performance of LAMP-Leish/HSP70 assay in comparison with disease duration ($\leq$ or $> 3$ months) or the parasite load ($\leq$ or $>$ than 10 parasites/µg of DNA tissue) was presented in Table 4. There was no significant difference in the positivity rate of LAMP-Leish/HSP70 assay in relation to the lesion's evolution time (p = 0.84) or parasite load (p = 0.54).

## Discussion

In the present study, the development and accuracy analysis of LAMP-Leish/HSP70 assay was performed for the molecular diagnosis of CL. This is the first LAMP protocol described for the detection of the HSP70 *Leishmania* gene. Nucleic acid tests (NATs), such as PCR, have a high precision rate for the laboratory diagnosis of CL, with pooled *Se* and *Sp* of 95% and 91%, respectively [40]. However, PCR remain confined to reference laboratories due to trained personnel and the high cost of precision instrumentation. LAMP assays are a viable alternative to

**Table 3. Sensitivity, specificity, and accuracy rates of LAMP-Leish/HSP70 assay and parasitological tests for CL diagnosis in biopsy samples.**

| Tests | Results: | | | | Sensitivity (%) (95% CI) | Specificity (%) (95% CI) | Accuracy (%) (95% CI) |
|---|---|---|---|---|---|---|---|
| | TP[a]* | FN[b]* | TN[c]* | FP[d]* | | | |
| LAMP-Leish/HSP70 assay | 52 | 8 | 40 | 0 | 86.7[e,f] (75.4–94.1) | 100 (91.2–100) | 92 (84.8–96.5) |
| Direct Microscopy | 13 | 47 | 40 | 0 | 21.7[e] (12.1–34.2) | 100 (91.2–100) | 53 (42.8–63.1) |
| Culture | 29 | 31 | 40 | 0 | 48.3[f] (35.2–61.6) | 100 (91.2–100) | 69 (59–77.9) |

*Gold standard test: kDNA-qPCR.

[a]True positive

[b]False negative

[c] True negative

[d]False positive.

McNemar test: The same letters

[e,f] indicate statistical differences between pairs of tests: e (LAMP-Leish/HSP70 assay x direct microscopy); f (LAMP-Leish/HSP70 assay x culture). p-value: p<0.0001.

**Table 4. Analysis of influence of the lesion evolution time and parasite load (calculated using kDNA qPCR) in CL cases about the performance of LAMP-Leish/ HSP70 assay.**

| LAMP assay | Evolution time of lesion | | Parasite load | |
|---|---|---|---|---|
| | ≤ 3 months (n = 28) % (95% CI) | >3 months (n = 32) % (95% CI) | low (n = 43) % (95% CI) | high (n = 17) % (95% CI) |
| LAMP-Leish/HSP70 | 85.7 (67.3–96)[a] | 87.5 (71–96.5)[a] | 88.4 (74.9–96.1)[b] | 82.4 (56.6–96.2)[b] |

LAMP-*Leish*/HSP70

[a] ($p$ = 0.84)

[b] ($p$ = 0.54).

overcome PCR limitations. The main advantage of LAMP is to amplify nucleic acid under isothermal amplification with high sensitivity and specificity [41,42].

The LOD for the developed LAMP assay was 100 fg. Comparatively, one *Leishmania* parasite is equivalent to 83.15 fg of DNA [37]. These LOD results using purified DNA are consistent with those of other LAMP studies: 20 fg of *L. major* and 200 fg of *L. tropica* using cysteine protease B gene [22]; 1 fg of *Leishmania* sp. using 18S primers [43]; 1 fg of *L. tropica* using kDNA primers [44]. The range of the LODs of LAMPs protocols for *Leishmania* can be influenced by the number of copies of the target gene, the species of *Leishmania* used in testing, and the readout methods for LAMP results [45]. The LAMP-Leish/HSP70 assay exhibited specificity for *Leishmania* spp. The analytical specificity of the LAMP-Leish/HSP70 assay is an advantage over other LAMP protocols, as some authors have reported nonspecific amplification using LAMP primers for the 18S region: *T. cruzi* and *T. brucei* DNA [28], and *T. evansi* DNA [27].

The LAMP-Leish/HSP70 assay exhibited good clinical performance, with a *Se* of 86.7%, *Sp* of 100%, and *Acc* of 92% for detecting *Leishmania* DNA in skin biopsy samples. The diagnostic *Acc* is consistent with previous studies regarding the use of LAMP protocols on biopsy samples for CL molecular diagnosis: 91.4% [29] 88,9% [22]; and 100% [46]. Other authors, using noninvasive samples and PCR as the reference standard for CL diagnosis, have reported *Se* ranging from 55.6% to 97% and *Sp* ranging from 91.7 to 100% [12,29,44,47]. It is important to highlight that there are still relatively few studies evaluating the *Acc* of LAMP assays for CL diagnosis using samples from Latin American patients: Brazil, with *Se* = 88.9% and *Sp* = 95.5% [29], and Peru, with *Se* = 88.9% and without data for *Sp*, [47], both using 18S primers; Suriname, with *Se* = 91.4–98% and without data for *Sp*. [28], and Colombia, with *Se* = 90.9–95% and *Sp* = 86%, both using 18S and kDNA primers LAMP [23,24]. Recently, the Loopamp™ *Leishmania* Detection kit, which targets the 18SrRNA gene and kDNA minicircles, was developed by Eiken Chemical Co. and FIND. This kit was evaluated for CL diagnosis in two countries with reported sensitivity and specificity rates of 91.4% and 91.7% in Suriname [12], and 87.6% and 70.6% in Afghanistan [13], respectively.

The use of SYBR green I dye on the inner side of the tube cover was evaluated for molecular diagnosis of visceral leishmaniasis (VL) [31]. The SYBR green I dye is used for readout of the LAMP assays, but this reagent causes inhibition of the Bst DNA polymerase enzyme [31,48,49]. This strategy avoids inhibition of the LAMP assay, as the SYBR green I dye contact with the other reagents only occurs at the end of reaction. Furthermore, carry-over contamination can be eliminated since the tube's cover can be kept closed [31].

There was no association between the positivity rate of LAMP-Leish/HSP70 assay with the evolution time of lesion, and parasite load in CL patients. The false negative results with LAMP-Leish/HSP70 may have been caused by differences in copy number between HSP70 gene (1–15 copies / parasite), and kDNA (> 10,000 copies / parasite) used as the gold standard in

this study [36,50]. The number of copies of these genes may vary depending on the *Leishmania* species. In the present study, it was not possible to determine the *Leishmania* species causing CL in the analyzed population, although *L. braziliensis* is the predominant species in the state of Minas Gerais, Brazil, where this study was carried out [51]. Using other isothermal technique amplification (recombinase polymerase amplification–RPA) with primers and probe for kDNA, the authors found similar results for molecular diagnosis of CL in Peru. *L. braziliensis* was highly prevalent (97.2%) in the samples isolated [52]. These results are important, since direct microscopy is less sensitive when the lesion evolution time exceeded 3 months [53,54]. In this study, the microscopy test was positive in only 13 out of 60 patients with CL. The LAMP-Leish/HSP70 assay correctly identified four times more cases compared to microscopy. The use of the HSP70-LAMP assay could lead to a reduction in the number of any PCR tests at reference centers, consequently shortening the time between initial care and the establishment of treatment for CL. Furthermore, the high specificity of the LAMP-Leish/HSP70 assay is an important characteristic, as the unjustified prescription of pentavalent antimonies or amphotericin B for patients without CL can be extremely harmful [17,55,56].

The LAMP-Leish/HSP70 assay was developed and successfully validated in a biopsy panel-based study for CL. However, there were limitations of this study that included the lack of identification of *Leishmania* species in the biopsy samples, the lack of assessment of the accuracy of LAMP-Leish/HSP70 assay using non-invasive samples along with simple method strategies for nucleic acid extraction (e.g. direct boiling). LAMP is a rapid and a user-friendly nucleic acid test, but still don't fulfill the REASSURED criteria [57]. Although some LAMP assays are currently being tested in resource-limited settings, it cannot be considered a true POC test due to its reliance on electricity, trained personnel and equipment [58]. It is essential to overcome the limitations mentioned above to be able to use the LAMP-Leish/HSP70 assay in the diagnosis of CL in resource-limited settings.

## Conclusion

The closed-tube SYBR green I LAMP-Leish/HSP70 assay is simple, highly accurate, and has potential to be used in resource-limited settings for CL diagnosis. Further validation of LAMP-Leish/HSP70 assay in prospective studies, including samples from minimally invasive methods, it's necessary to confirm sensitivity and specificity values compared with other available LAMP assays, and the main tests used in routine diagnosis of CL. Although the LAMP assay presented here has been evaluated for the molecular diagnosis of CL, it should also be evaluated for the diagnosis of ML, MCL and VL.

## Supporting information

**S1 Table. Demographic and clinical characteristics of all participant patients and the results of diagnostic tests.**
(XLSX)

**S1 Raw images.**
(PDF)

## Acknowledgments

The authors would like to thank the Leishmaniasis Reference Center (LRC-FIOCRUZ/MG).

## Author Contributions

**Conceptualization:** Arthur Ribeiro Cheloni Soares, Daniel Moreira de Avelar.

**Data curation:** Arthur Ribeiro Cheloni Soares, Daniel Moreira de Avelar.

**Formal analysis:** Arthur Ribeiro Cheloni Soares, Verônica Cardoso Santos de Faria, Daniel Moreira de Avelar.

**Funding acquisition:** Daniel Moreira de Avelar.

**Investigation:** Arthur Ribeiro Cheloni Soares, Verônica Cardoso Santos de Faria.

**Methodology:** Arthur Ribeiro Cheloni Soares, Verônica Cardoso Santos de Faria.

**Project administration:** Daniel Moreira de Avelar.

**Supervision:** Daniel Moreira de Avelar.

**Validation:** Daniel Moreira de Avelar.

**Writing – original draft:** Arthur Ribeiro Cheloni Soares, Verônica Cardoso Santos de Faria, Daniel Moreira de Avelar.

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
