## [Decision Letter · Decision Letter 0]

10 Apr 2024

PONE-D-24-03257Development and accuracy evaluation of loop-mediated isothermal amplification assays in diagnosing cutaneous leishmaniasisPLOS ONE

Dear Dr. Avelar,

Thank you for submitting your manuscript to PLOS ONE. After careful consideration, we feel that it has merit but does not fully meet PLOS ONE’s publication criteria as it currently stands. Therefore, we invite you to submit a revised version of the manuscript that addresses the points raised during the review process. Please take into account the comments provided by the reviewers as well as:

-Present the study's context (background) within the abstract.

-In the introduction and in supporting the discussion on the accuracy of existing diagnostic techniques, it is important to reference systematic reviews with meta-analyses that have already been published on accuracy of diagnostic tests for tegumentary leishmaniasis.

-Enhance the explanation of the sample calculation to ensure reproducibility.

-Elaborate on the practical application of the technique, discuss the limitations of the study, and suggest avenues for future research in the discussion section.

-Mention all supplementary materials in the body of the text.

-Clearly inform where the data can be found, as they were not made available as supplementary material.

-Standardize the article according to the recommendations outlined by PLoS One.

We look forward to receiving your revised manuscript.

Kind regards,

Vinícius Silva Belo

Academic Editor

PLOS ONE

Journal Requirements:

   "This study was supported by and The Research Foundation of the State of Minas Gerais (FAPEMIG - APQ-00802-20). ARCS would also like to thank FAPEMIG-Brazil for its scholarship grant."

Reviewers' comments:

Reviewer's Responses to Questions

**Comments to the Author**

1. Is the manuscript technically sound, and do the data support the conclusions?

Reviewer #1: Yes

Reviewer #2: Yes

2. Has the statistical analysis been performed appropriately and rigorously? 

Reviewer #1: N/A

Reviewer #2: Yes

3. Have the authors made all data underlying the findings in their manuscript fully available?

Reviewer #1: Yes

Reviewer #2: Yes

4. Is the manuscript presented in an intelligible fashion and written in standard English?

Reviewer #1: Yes

Reviewer #2: Yes

5. Review Comments to the Author

Reviewer #1: The submitted paper is well driven and composed. However, there are some incongruences that need clarification.

Comments:

-Line 27: what species do you want to amplify?, CL is caused by several species and that is important for the specificity and the design of the LAMP assay. Some epidemiology data is lacked here. This would explain why you use conserved regions for the design (line 101).

-Line 67: indicate the clinical sensitivity of CL Detect™ Rapid Test for comparison.

-Line 70: It is lack of data regarding sensibility and specificity of molecular techniques such as PCR. Please, describe it. It is important to know the accuracy of molecular data – this is why a LAMP amplification is pursued in this work.

-Line 83: why did you design a new LAMP assay instead of using a previously standardized LAMP technique?

-Line 93: highlight the advantages of choosing these targets for this new LAMP assay.

-Line 94: I would mention that you tried to standardize 18S-LAMP assay without success and remove this assay from the draft due that there are other protocols already working good (León et al, 2018; Ghodrati et al, 2017; Tiwananthagorn et al., 2017; etc..). There is no point in its description, once you have not used it for diagnosis, main objective of the manuscript.

-Line 281: what are the sensitivity rates presented by the reference q-PCR and direct microscopy examination and aspirate of culture?, similar to LAMP?, introduce this information in Table 3.

-Line 292: data from q-PCR are lacking. What was the cut-off (line 186)?. Where those negatives by LAMP (FN) the ones with lower parasitic load by kDNA-qPCR?, is a problem of sensibility?. Do you know the specie causing the biopsies?, is it a problem of specificity regarding some Leishmania species?

-Line 297: why did you use 10 parasites/µg of DNA tissue?.

Please for comparison describe accuracy of all the cited references in Discussion.

-Line 319: it is described the LOD of purified Leishmania DNA which is different from Leishmania DNA in biopsies (they include human DNA). Please verify if the LOD is from purified DNA or biopsies in the references added, and insert a range of LOD from these references in the text.

-Line 334: insert the performance of the cited references in the text as a range.

-Line 336: why is this described apart regarding line 334?, My advice is to cite here only those with PCR as a reference (microscopy results has a poorer relationship). This will help to compare with the protocol optimized here. Are there any study including a PCR protocol using a HSP70 target?.

-Line 341: please describe the results of these studies for comparison. Other places out of Latin American countries?, which is the target? is a conserved target among species?.

-Line 351: describe target, specie..

-Line 354: any PCR, correct it.

-Line 355: what is the advantage regarding the other LAMP protocols?

-Line 357: 357: as other LAMP protocols?

-Line 361: did your LAMP protocol suffer from inhibition?, did you add SYBR green to avoid it?, all LAMP protocols that not use SYBR green are prompt to be inhibited? Because you have cited several studies in your discussion with good results.

-Line 374: why did it appear here mucosal leishmaniasis?, not described before its importance. If so, why did not do it in your study?

Reviewer #2: This in an interesting manuscript since an innovative and easy-to-perform test for cutaneous leishmaniasis is highly needed.

References are not correctly cited throughout the text according to the rules of PLOS ONE: “References are listed at the end of the manuscript and numbered in the order that they appear in the text. In the text, cite the reference number in square brackets (…)”, see: https://journals.plos.org/plosone/s/submission-guidelines#loc-references

Introduction

Line 44: The authors can cite more actualized data: WHO, 2022 with the link https://apps.who.int/neglected_diseases/ntddata/leishmaniasis/leishmaniasis.html in the references, because the 90 endemic countries in the world did not change from 2021 to 2022, when last updated by WHO.

Lines 47-49: “CL is the most common clinical form, and unsolved CL cases can progress to mucosal disease, diffuse CL, or disseminated CL (GIANCHECCHI and MONTOMOLI, 2020).” This is not entirely true, because most CL do not evolve into any other forms of the disease. Mucosal leishmaniasis affects only 5-6% of the Brazilian cases of tegumentary leishmaniasis reported to the Brazilian Ministry of Health and is mainly related to areas where Leishmania braziliensis is endemic. It is much rarer with other species of Leishmania especially in other parts of the world outside American continent. Diffuse CL is associated with Leishmania amazonensis in rare patients with anergy towards this species, and it is much more uncommom than localized cutaneous leishmaniasis caused by this same species of Leishmania. Disseminated leishmaniasis is usually uncommon, occurs in Brazil mainly in endemic areas for Leishmania braziliensis and may or may not be associated with immunosuppression; other species of Leishmania can be implicated in this unusual presentation of the disease. The cited reference is also not adequate to discuss the different clinical forms of American tegumentary leishmaniasis, since is refers to Italy. I suggest the authors to search for a more adequate reference when referring to the different clinical presentations of the disease, preferably a Brazilian reference in English or even WHO technical report series 949, Control of leishmaniasis, 2010.

MATERIALS AND METHODS

Alignment of Leishmania spp genome sequences and primer design

Lines 98-106: In the first use of an acronym, please specify it full-length, particularly F3, B3, FIP and BIP, although they are defined after in the legend of Table 1.

Preparation of DNA reference

Lines 109-110: in the first use of the subgenus, use it full-length (Leishmania (Leishmania) amazonensis; Leismania (Viannia) braziliensis), then you can use L (L.) donovani, L (V.) guyanensis and so on.

RESULTS

The first column of Table 1 must be aligned to the left. (The same applies to Table 2).

Legend of Table 1

Lines 219-220: I was not able to detect FLP and BLP in the table. Please define “pb” in the legend.

Please verify the words in the top of Figure 1, they are not legible. The figures are also not distinct, and the difference between orange and greenish yellow colors are not so clear.

The same occurred in Figure 2.

In Figure 3, the photographs are more distinct.

Table 3

Lines 290-291: Specify the letters in “The same letters (…)”

Line 301: Since the Figures and Tables must be self-explanatory, specify in the title of Table 4 that the parasite load was calculated using kDNA-qPCR.

Discussion

Lines 320-321: “Comparatively, JARA et al. (2013) established that 83.15 fg of Leishmania DNA is equivalent to one parasite equivalent.” Is this sentence correct? Please verify.

Lines 342-343: It is worthy to specify the gene targets used in the Loopamp® kit.

It is important to add a few sentences to discuss the limitations of the study before the Conclusions. In these sentences, I suggest the authors discuss if a molecular method such as LAMP, although easier to perform when compared to the gold standard which is PCR, could be considered as a test for using in the point-of-care or if it does not yet fulfill the requirements for a point-of-care test and specify why not.

Please be certain that all the references follow the rules of PLOS ONE, since there are different presentations of some of them.

6. PLOS authors have the option to publish the peer review history of their article (what does this mean?). If published, this will include your full peer review and any attached files.

Reviewer #1: **Yes: **Eugenia Carrillo Gallego

Reviewer #2: No

---

## [Author Response · Author response to Decision Letter 0]

24 May 2024

We would like to thank the academic editor and reviewers for their careful review of the manuscript "Development and accuracy evaluation of a new loop-mediated isothermal amplification assay targeting the HSP70 gene for the diagnosis of cutaneous leishmaniasis" (PONE-D-24-03257). As suggested by reviewer 1, we decided to remove the 18S LAMP assay from the manuscript. 

Answer to Academic Editor 

1. Present the study's context (background) within the abstract.

This information was added. Lines 27-28. 

2. In the introduction and in supporting the discussion on the accuracy of existing diagnostic techniques, it is important to reference systematic reviews with meta-analyses that have already been published on accuracy of diagnostic tests for tegumentary leishmaniasis.

Meta-analyses were referenced in the Introduction and Discussion, for PCR (lines 69-70; 329) and LAMP studies (lines 80-81). Sensitivity and specificity data of this studies were added in the body of the text. 

3. Enhance the explanation of the sample calculation to ensure reproducibility.

This information was added. Lines 179-183.

4. Elaborate on the practical application of the technique, discuss the limitations of the study, and suggest avenues for future research in the discussion section.

This information was added. Lines 389-397. 

5. Mention all supplementary materials in the body of the text.

This information was added. Lines 145 and 287. 

6. Clearly inform where the data can be found, as they were not made available as supplementary material.

All relevant data are within the manuscript and its Supporting Information files. This information was added in lines 426-427.

7. Standardize the article according to the recommendations outlined by PLoS One.

Done.

Answer Letter to Reviewers

Reviewer #1: 

• Line 27: what species do you want to amplify?, CL is caused by several species and that is important for the specificity and the design of the LAMP assay. Some epidemiology data is lacked here. This would explain why you use conserved regions for the design (line 101).

We developed a pan-Leishmania assay able to amplify the main species on the subgenus Viannia and Leishmania responsible for causing cutaneous leishmaniasis. This information was added in lines 27-28. 

• Line 67: indicate the clinical sensitivity of CL Detect™ Rapid Test for comparison.

This information was added. Lines 65-68.

• Line 70: It is lack of data regarding sensibility and specificity of molecular techniques such as PCR. Please, describe it. It is important to know the accuracy of molecular data – this is why a LAMP amplification is pursued in this work.

Meta-analyses were referenced in the Introduction and Discussion, for PCR (lines 69-70; 329) and LAMP studies (lines 80-81). Sensitivity and specificity data of this studies were added in the body of the text. 

• Line 83: why did you design a new LAMP assay instead of using a previously standardized LAMP technique?

Our research group previously evaluated a LAMP assay (18S target; Adams et al., 2010) on samples collected from Brazilian patients (de Faria et al., 2022). This assay, although presenting high sensitivity and specificity, showed cross reaction with Trypanosoma cruzi DNA. Besides that, this is the first LAMP assay based on HSP70 gene for DNA detection of Leishmania. HSP70 gene was selected because it is commonly used for molecular diagnosis of Leishmaniasis, including Leishmania species identification. This information was added. Lines 93-97.

• Line 93: highlight the advantages of choosing these targets for this new LAMP assay.

This information was added. Lines 93-97.

• Line 94: I would mention that you tried to standardize 18S-LAMP assay without success and remove this assay from the draft due that there are other protocols already working good (León et al, 2018; Ghodrati et al, 2017; Tiwananthagorn et al., 2017; etc..). There is no point in its description, once you have not used it for diagnosis, main objective of the manuscript.

All mentions to the 18S LAMP assay were removed. 

• Line 281: what are the sensitivity rates presented by the reference q-PCR and direct microscopy examination and aspirate of culture?, similar to LAMP?, introduce this information in Table 3.

Being the reference standard, only samples positive by kDNA-qPCR test were considered cases of cutaneous leishmaniasis. This information is present in Table 3. Sensitivity and specificity of direct microscopy and culture of aspirate are also in Table 3. 

• Line 292: data from q-PCR are lacking. What was the cut-off (line 186)? qPCR criteria were added from line 199 to 207. Where those negatives by LAMP (FN) the ones with lower parasitic load by kDNA-qPCR?, is a problem of sensibility? No. The parasite load was not deemed statistically relevant to the positivity rate. More information was added in lines 370-374. We believe these results were due the differences in number of copies between HSP70 and kDNA genes. Do you know the specie causing the biopsies?, is it a problem of specificity regarding some Leishmania species?

Specie identification was not realized, but L. braziliensis is the predominant species in the State of Minas Gerais, Brazil, where this study was carried out. This limitation was cited in Discussion (added in lines 374-377 and 392).

• Line 297: why did you use 10 parasites/µg of DNA tissue?

The reference for the use of 10 parasites/µg of DNA tissue was added in manuscript. Lines 219-220.

• Please for comparison describe accuracy of all the cited references in Discussion.

Information was added in lines 330, 347, 349-361.

• Line 319: it is described the LOD of purified Leishmania DNA which is different from Leishmania DNA in biopsies (they include human DNA). Please verify if the LOD is from purified DNA or biopsies in the references added, and insert a range of LOD from these references in the text.

We kept in the text only data regarding LOD of purified Leishmania DNA. Information was added in lines 336-339.

• Line 334: insert the performance of the cited references in the text as a range.

Done. 

• Line 336: why is this described apart regarding line 334?, My advice is to cite here only those with PCR as a reference (microscopy results has a poorer relationship). This will help to compare with the protocol optimized here.

We kept only studies that used PCR as a reference. Lines 350-351.

• Are there any study including a PCR protocol using a HSP70 target?

Yes. The HSP70 target is commonly used for molecular diagnosis of leishmaniasis. This information was added in Introduction (lines 93-94). However, even in PCR comparative studies, its performance is lower than kDNA PCR. 

• Line 341: please describe the results of these studies for comparison. Other places out of Latin American countries?, which is the target? is a conserved target among species?.

This information was added. Lines 353-361. The targets were specified and all of them are conserved. 

• Line 351: describe target, specie..

Done.

• Line 354: any PCR, correct it.

Done.

• Line 355: what is the advantage regarding the other LAMP protocols?

The HSP70 LAMP assay have not shown cross reaction with other species of parasites outside Leishmania. Some assays standardized with 18S target presented unspecific amplification with Trypanosoma cruzi, T. brucei and T. evansi. This information was added in manuscript. Lines 343-345. 

• Line 357: 357: as other LAMP protocols?

Yes. Any lamp protocols can present these advantages. Sentence corrected in manuscript. Line: 384. 

• Line 361: did your LAMP protocol suffer from inhibition?, did you add SYBR green to avoid it?, all LAMP protocols that not use SYBR green are prompt to be inhibited? Because you have cited several studies in your discussion with good results.

No, SYBR green I was used as intercalating dye for visual analysis of amplified material. This reagent inhibits Bst DNA polymerase action and it is generally applied after the reaction is done, but this requires the tube to be opened, which may lead to the contamination of the workbench. It was added on the inner side of tube’s cover so it could be centrifuged down after the reaction is done. This information is presented in lines 362-368.

• Line 374: why did it appear here mucosal leishmaniasis?, not described before its importance. If so, why did not do it in your study?

Mucosal leishmaniasis appeared as a perspective for future accuracy studies. Its was not evaluated in this study because this clinical form requires its own study design, exclusively using ML samples, which were not in the scope of the present study. Besides that, we corrected the phrases regarding ML, as to include mucocutaneous and visceral leishmaniasis as perspectives (lines 406-407. 

Reviewer #2:

• References are not correctly cited throughout the text according to the rules of PLOS ONE: “References are listed at the end of the manuscript and numbered in the order that they appear in the text. In the text, cite the reference number in square brackets (…)”, see: https://journals.plos.org/plosone/s/submission-guidelines#loc-references

References were corrected. 

• Introduction

Line 44: The authors can cite more actualized data: WHO, 2022 with the link https://apps.who.int/neglected_diseases/ntddata/leishmaniasis/leishmaniasis.html in the references, because the 90 endemic countries in the world did not change from 2021 to 2022, when last updated by WHO.

The suggestion was accepted. Line: 45

• Lines 47-49: “CL is the most common clinical form, and unsolved CL cases can progress to mucosal disease, diffuse CL, or disseminated CL (GIANCHECCHI and MONTOMOLI, 2020).” This is not entirely true, because most CL do not evolve into any other forms of the disease. Mucosal leishmaniasis affects only 5-6% of the Brazilian cases of tegumentary leishmaniasis reported to the Brazilian Ministry of Health and is mainly related to areas where Leishmania braziliensis is endemic. It is much rarer with other species of Leishmania especially in other parts of the world outside American continent. Diffuse CL is associated with Leishmania amazonensis in rare patients with anergy towards this species, and it is much more uncommom than localized cutaneous leishmaniasis caused by this same species of Leishmania. Disseminated leishmaniasis is usually uncommon, occurs in Brazil mainly in endemic areas for Leishmania braziliensis and may or may not be associated with immunosuppression; other species of Leishmania can be implicated in this unusual presentation of the disease. The cited reference is also not adequate to discuss the different clinical forms of American tegumentary leishmaniasis, since is refers to Italy. I suggest the authors to search for a more adequate reference when referring to the different clinical presentations of the disease, preferably a Brazilian reference in English or even WHO technical report series 949, Control of leishmaniasis, 2010.

The suggestion was accepted and we used this exact technical report. Line: 48. 

• MATERIALS AND METHODS

Alignment of Leishmania spp genome sequences and primer design

Lines 98-106: In the first use of an acronym, please specify it full-length, particularly F3, B3, FIP and BIP, although they are defined after in the legend of Table 1.

Acronyms specified. 

• Preparation of DNA reference

Lines 109-110: in the first use of the subgenus, use it full-length (Leishmania (Leishmania) amazonensis; Leismania (Viannia) braziliensis), then you can use L (L.) donovani, L (V.) guyanensis and so on.

Done.

• RESULTS

The first column of Table 1 must be aligned to the left. (The same applies to Table 2).

Legend of Table 1

Lines 219-220: I was not able to detect FLP and BLP in the table. Please define “pb” in the legend.

Done.

• Please verify the words in the top of Figure 1, they are not legible. The figures are also not distinct, and the difference between orange and greenish yellow colors are not so clear.

The same occurred in Figure 2.

In Figure 3, the photographs are more distinct.

Figures were corrected. 

• Table 3

Lines 290-291: Specify the letters in “The same letters (…)”

Letters specified. 

Line 301: Since the Figures and Tables must be self-explanatory, specify in the title of Table 4 that the parasite load was calculated using kDNA-qPCR.

Information added.

• Discussion

Lines 320-321: “Comparatively, JARA et al. (2013) established that 83.15 fg of Leishmania DNA is equivalent to one parasite equivalent.” Is this sentence correct? Please verify.

Sentence corrected. Lines: 335-336.

• Lines 342-343: It is worthy to specify the gene targets used in the Loopamp® kit.

Information added. Lines: 357-359.

• It is important to add a few sentences to discuss the limitations of the study before the Conclusions. In these sentences, I suggest the authors discuss if a molecular method such as LAMP, although easier to perform when compared to the gold standard which is PCR, could be considered as a test for using in the point-of-care or if it does not yet fulfill the requirements for a point-of-care test and specify why not.

Thanks for the comments. We added information regarding the limitations of the present study and LAMP in general. LAMP assays cannot be considered POC tests, but it is possible to implement molecular assays as near POC tests. This information is presented in lines 390-398.

---

## [Decision Letter · Decision Letter 1]

17 Jun 2024

PONE-D-24-03257R1Development and accuracy evaluation of a new loop-mediated isothermal amplification assay targeting the HSP70 gene for the diagnosis of cutaneous leishmaniasisPLOS ONE

Dear Dr. Avelar,

Thank you for submitting your manuscript to PLOS ONE. After careful consideration, we feel that it has merit but does not fully meet PLOS ONE’s publication criteria as it currently stands. Therefore, we invite you to submit a revised version of the manuscript that addresses the points raised during the review process.

We look forward to receiving your revised manuscript.

Kind regards,

Vinícius Silva Belo

Academic Editor

PLOS ONE

Journal Requirements:

Reviewers' comments:

Reviewer's Responses to Questions

**Comments to the Author**

1. If the authors have adequately addressed your comments raised in a previous round of review and you feel that this manuscript is now acceptable for publication, you may indicate that here to bypass the “Comments to the Author” section, enter your conflict of interest statement in the “Confidential to Editor” section, and submit your "Accept" recommendation.

Reviewer #1: All comments have been addressed

Reviewer #2: (No Response)

2. Is the manuscript technically sound, and do the data support the conclusions?

Reviewer #1: Yes

Reviewer #2: Yes

3. Has the statistical analysis been performed appropriately and rigorously? 

Reviewer #1: Yes

Reviewer #2: Yes

4. Have the authors made all data underlying the findings in their manuscript fully available?

Reviewer #1: Yes

Reviewer #2: Yes

5. Is the manuscript presented in an intelligible fashion and written in standard English?

Reviewer #1: Yes

Reviewer #2: Yes

6. Review Comments to the Author

Reviewer #1: All my comments have been properly addressed but the following corrections will improve the paper:

• Line 47: remove “as”

• Line 67: indicate the clinical sensitivity of CL Detect™ Rapid Test for comparison.

Rephrase the sentence in line 65: “been evaluated in some endemic countries (31.3%-65.4% sensitivity when compared to microscopy and PCR) [12-14].

• Line 82: delete “.” after respectively

• Line 84: The previous 18S cross-reactivity with Trypanosoma cruzi DNA should be mentioned (de Faria et al., 2022).

• Line 92: highlight in a sentence the advantages of choosing HSP70 for this new LAMP assay.

- most common target used for Leishmania species identification,

- presents conserved and polymorphic regions: being able to differentiate a wide range of species from different geographical origins, especially the ATL species,

- Etc

• Line 93: Rephrase the sentence:

The target was selected because is commonly used for molecular diagnosis of Leishmaniasis, including Leishmania species identification. Besides that, this is the first LAMP assay based on HSP70 gene for DNA detection of Leishmania. The accuracy of LAMP- Leish/HSP70 assay was evaluated for the diagnosis of CL.

The target was selected because is commonly used for molecular diagnosis of Leishmaniasis, including Leishmania species identification. For the first time, the accuracy of LAMP- Leish/HSP70 assay to diagnose CL was evaluated in this study.

• Line 167: The recruitment period occurred from 2017 to 2019.

• Line 307: : delete “)” here e;f)

• Line 310-12: Please indicate where these data are in the paper. If not, add median values and IQR to the text:

Recent lesions (median of 90 days) had a significantly higher parasite load compared to lesions with a longer evolution time (median of 120 days) (U Mann-Whitney test, p = 0.03).

• Discussion: Each of the available PoC diagnostic methods has a limitation in at least one REASSURED criterion. It is very difficult to create pathogen-specific PoC diagnostic platforms that meet all of these criteria. In limitations, the authors should cite that Loop-mediated isothermal amplification (LAMP) is a rapid, user friendly isothermal nucleic acid amplification method that has almost all ASSURED properties. They might include some information from https://doi.org/10.1080/14737159.2021.1873769.

• Write consistently one of each term along the text: hsp70-LAMP assay or HSP70-LAMP assay or LAMP-Leish/HSP70 assay

• Figure 1. Please, improve the typewriting, it is unreadable

Reviewer #2: The authors made the suggested corrections, but a few minor changes are still needed:

Lines 169-171: This sentence must be removed: “All individuals participated voluntarily in the study and informed consent was obtained from each participant of from their legal guardian for underage participants”, because it repeats information already stated in lines 165-167.

Table 1, first line, change “pb” into “bp” to be standardized with the legend.

Line 363: “(…) in the inner side of the tube ever (…)” must be replaced by: “(…) in the inner side of the tube cover (…)”

Line 420: “This study was supported by and The Research Foundation (…)” must be replaced by: “This study was supported by The Research Foundation (…)”

Reference 16 must be adapted to Vancouver style.

7. PLOS authors have the option to publish the peer review history of their article (what does this mean?). If published, this will include your full peer review and any attached files.

Reviewer #1: **Yes: **Eugenia Carrillo Gallego

Reviewer #2: No

---

## [Author Response · Author response to Decision Letter 1]

24 Jun 2024

Reviewer #1: 

• Line 47: remove “as”

 Done. 

• Line 67: indicate the clinical sensitivity of CL Detect™ Rapid Test for comparison.

Rephrase the sentence in line 65: “been evaluated in some endemic countries (31.3%-65.4% sensitivity when compared to microscopy and PCR) [12-14].

 This information was added.

• Line 82: delete “.” after respectively

 Done. 

• Line 84: The previous 18S cross-reactivity with Trypanosoma cruzi DNA should be mentioned (de Faria et al., 2022).

 This information was added, as mentioned in reference 28.

• Line 92: highlight in a sentence the advantages of choosing HSP70 for this new LAMP assay.

- most common target used for Leishmania species identification,

- presents conserved and polymorphic regions: being able to differentiate a wide range of species from different geographical origins, especially the ATL species,

- Etc

 Sentence was rewritten with the information added. Lines: 96-99.

• Line 93: Rephrase the sentence:

The target was selected because is commonly used for molecular diagnosis of Leishmaniasis, including Leishmania species identification. Besides that, this is the first LAMP assay based on HSP70 gene for DNA detection of Leishmania. The accuracy of LAMP- Leish/HSP70 assay was evaluated for the diagnosis of CL.

The target was selected because is commonly used for molecular diagnosis of Leishmaniasis, including Leishmania species identification. For the first time, the accuracy of LAMP- Leish/HSP70 assay to diagnose CL was evaluated in this study.

 Sentence was rewritten with the information added. Lines: 96-99. 

• Line 167: The recruitment period occurred from 2017 to 2019.

 Done.

• Line 307: : delete “)” here e;f)

 Done. 

• Line 310-12: Please indicate where these data are in the paper. If not, add median values and IQR to the text:

Recent lesions (median of 90 days) had a significantly higher parasite load compared to lesions with a longer evolution time (median of 120 days) (U Mann-Whitney test, p = 0.03).

 The values were added, and analysis was remade. There was a typing mistake in the p value (it was 0.03 and the correct one is 0.3), that was corrected and the interpretation was rewritten. The raw data was referenced. Lines: 315-320.

• Discussion: Each of the available PoC diagnostic methods has a limitation in at least one REASSURED criterion. It is very difficult to create pathogen-specific PoC diagnostic platforms that meet all of these criteria. In limitations, the authors should cite that Loop-mediated isothermal amplification (LAMP) is a rapid, user friendly isothermal nucleic acid amplification method that has almost all ASSURED properties. They might include some information from https://doi.org/10.1080/14737159.2021.1873769.

 Thanks for the suggestion. This information and the reference were added. Lines: 404-409. 

• Write consistently one of each term along the text: hsp70-LAMP assay or HSP70-LAMP assay or LAMP-Leish/HSP70 assay

 Done. We kept “LAMP-Leish/HSP70 assay” in the text. 

• Figure 1. Please, improve the typewriting, it is unreadable

 The typewriting is unreadable due to compression of the file made available in the PLOS editorial manager platform. The uncompressed version of all images, uploaded following PLOS guidelines, are present in the editorial manager, under "File Inventory" link. 

Reviewer #2:

• Lines 169-171: This sentence must be removed: “All individuals participated voluntarily in the study and informed consent was obtained from each participant of from their legal guardian for underage participants”, because it repeats information already stated in lines 165-167.

The sentence was removed. 

• Table 1, first line, change “pb” into “bp” to be standardized with the legend. 

Done. 

• Line 363: “(…) in the inner side of the tube ever (…)” must be replaced by: “(…) in the inner side of the tube cover (…)”

Done.

• Line 420: “This study was supported by and The Research Foundation (…)” must be replaced by: “This study was supported by The Research Foundation (…)”

Done.

• Reference 16 must be adapted to Vancouver style.

Reference added.

---

## [Editor Report · Decision Letter 2]

27 Jun 2024

Development and accuracy evaluation of a new loop-mediated isothermal amplification assay targeting the HSP70 gene for the diagnosis of cutaneous leishmaniasis

PONE-D-24-03257R2

Dear Dr. Avelar,

We’re pleased to inform you that your manuscript has been judged scientifically suitable for publication and will be formally accepted for publication once it meets all outstanding technical requirements.

Kind regards,

Vinícius Silva Belo

Academic Editor

PLOS ONE
---

## [Editor Report · Acceptance letter]

13 Aug 2024

PONE-D-24-03257R2 

PLOS ONE

Dear Dr. Avelar, 

I'm pleased to inform you that your manuscript has been deemed suitable for publication in PLOS ONE. Congratulations! Your manuscript is now being handed over to our production team.

Kind regards, 

on behalf of

Dr. Vinícius Silva Belo 

Academic Editor

PLOS ONE